# The Effect of Resveratrol on Blood Lipid Profile: A Dose-Response Meta-Analysis of Randomized Controlled Trials

**DOI:** 10.3390/nu14183755

**Published:** 2022-09-11

**Authors:** Xinyi Cao, Wang Liao, Hui Xia, Shaokang Wang, Guiju Sun

**Affiliations:** Key Laboratory of Environmental Medicine and Engineering of Ministry of Education, Department of Nutrition and Food Hygiene, School of Public Health, Southeast University, Nanjing 210009, China

**Keywords:** resveratrol, lipid profile, randomized controlled trial, dose-response, meta-analysis

## Abstract

(1) Background: The effects of resveratrol on blood lipids are controversial. Whether there is a dose-response of the lipid profile upon resveratrol supplementation is unknown. (2) Methods: This dose-response meta-analysis of randomized controlled trials (RCTs) was performed to explore the effects of resveratrol supplementation on lipid profile. A systematical and comprehensive search of several databases was conducted by 30 June 2022. (3) Results: The results indicated that the intake of resveratrol could significantly decrease the total cholesterol (TC) (mean difference = −10.28; 95%CI: −13.79, −6.76, *p* < 0.001), triglyceride (TG) (Mean difference = −856; 95%CI: −12.37, −4.75, *p* < 0.001) and low-density lipoprotein cholesterol (LDL-C) (mean difference = −5.69; 95%CI: −11.07, −0.31, *p* = 0.038) level, but did not alter the level of high-density lipoprotein cholesterol (HDL-C). In the non-linear dose–response analysis, we observed a significant effect of the supplementation dosage on the level of LDL-C (p-nonlinearity = 0.002). Results from the sub-group analysis showed that the reduction of LDL-C was more significant in the trials with a duration of ≥12 weeks and in subjects with type 2 diabetes mellitus. (4) Conclusion: Findings from this study suggest that resveratrol may be beneficial to reduce TC, TG, and LDL-C levels in the blood. The dosage of the resveratrol intervention is an essential factor that affects the level of LDL-C.

## 1. Introduction

Cardiovascular disease (CVD) is the leading cause of morbidity and mortality worldwide, and its prevalence is increasing. Dyslipidemia is clinically manifested as abnormally high blood lipid concentration, including the elevations of total cholesterol and low-density lipoprotein cholesterol (LDL-C), which is a major risk factor for CVD [1,2]. Due to the side effects of pharmaceutic drugs, many patients prefer a non-drug treatment [3]. The 2016 EU Guidelines on dyslipidemia are also appealing to the public to reinforce the mainstream importance of nonpharmacological treatment [4].

Resveratrol (3,5,4′-trihydroxystilbene), a nonflavonoid polyphenol organic compound, is an antitoxin produced by various plants, such as grapes, blueberries, raspberries, mulberries, peanuts, and knotweed [5,6,7]. Structurally, resveratrol is composed of a double bond with two phenolic rings at each end, which can be isomerized *cis* or *trans*. Of note, resveratrol has been reported to have health-promoting effects, such as anti-inflammatory, anti-cancer, anti-apoptotic, anti-osteoporosis, and antioxidant activities [8,9,10,11,12]. Resveratrol has also been shown to protect against vascular aging by modulating the renin-angiotensin system [13] as well as ameliorating endothelial dysfunction by activating SIRT1 [14]. Moreover, supplementation of resveratrol could contribute to the upregulation of transthyroxine, which has beneficial effects on the pathology of Alzheimer’s disease [15].

Importantly, the potential roles of resveratrol in regulating lipids have been implicated. The browning of excessive white adipose tissue could be inhibited by resveratrol, suggesting the anti-obesity effect of resveratrol [16]. In addition, the formation of the foam cell could be suppressed by resveratrol to prevent dynamic atherosclerosis in the initial stage [17]. Moreover, the gene expression profile related to hepatic lipid metabolism involved in the modulation of dyslipidemia could be altered by resveratrol [18].

Although a number of studies have reported the effects of resveratrol on blood lipids in humans, the results are inconsistent. It has been reported that the intervention of resveratrol could significantly affect the levels of TC, TG, HDL-C, and LDL-C [19,20,21,22,23]. On the contrary, results from some studies showed insignificant effects [24,25,26,27]. In addition, several meta-analyses on the effect of resveratrol on blood lipids have been published, but the correlation between the dose of resveratrol and the response of blood lipids is unavailable [2,28,29,30,31].On the other hand, several randomized controlled trials with higher doses of resveratrol and longer duration of intervention to find the effects of resveratrol on lipid parameters have been published in recent years, which showed new evidence [24,32]. Also, trials have shown that consuming 5000 mg/day did not cause significant side effects [14].

Given the potential role of resveratrol in regulating dyslipidemia and the inconsistent findings from various clinical trials, we integrated and updated various studies on the effects of resveratrol on lipids, and added the dose-response relationship on the effects of resveratrol on lipids, which aimed to provide a more comprehensive view of the regulation of lipids by resveratrol.

## 2. Materials and Methods

Our study was designed in accordance with the 2020 Systematic Evaluation and Meta-analysis (PRISMA) Statement of Preferred Reporting Program Guidelines.

### 2.1. Literature Search Strategy

We systematically searched PubMed, Web of Science, Embase, and Cochrane library databases to identify relevant articles published before 30 June 2022. Literature written in English was included as candidates. The search used the terms in titles or abstracts combined with MeSH. The search string is shown blew: (“Resveratrol”) and (“Cholesterol” OR “HDL” OR “LDL” OR “Triglycerides” OR “Li-poprotein” OR “low-density lipoprotein” OR “LDL-C” OR “VLDL” OR “HDL-C” OR “TG” OR “TC”). We also reviewed the reference list of each systematic review and meta-analysis to identify any relevant studies that might have been ignored.

### 2.2. Inclusion and Exclusion Criteria

The trial was selected according to the following criteria: (i) the study should be a clinical trial; (ii) the study should be designed as a randomized controlled trial to study the effects of resveratrol on blood lipids; (iii) at least of one of the TC, LDL-C, HDL-C or TG was used as the outcome measure; (iv) no other foods or supplements were used in the intervention and control groups. The trial was excluded if it meets one or more of the following criteria: (i) using any intervention in the control group; (ii) using other food supplements with resveratrol; (iii) not including the control group; (iv) with other lifestyle modifications, such as physical exercise.

### 2.3. Data Extraction

Two independent researchers evaluated and extracted data from articles that met the criteria by a specially designed data collection form. Extracted data includes: the last name of the first author, publication year, country, age range, male-female ratio, the sample size in each group, dose (mg/d), the follow-up period, and study design. In addition, standard deviation, net mean change of TC, TG, LDL-C, and HDL-C, and weight mean difference in each literature were also extracted.

### 2.4. Quality Assessment

The Cochrane Collaboration risk of bias tool was applied to evaluate the quality of selected literature using the following domains: “random sequence generation, allocation concealment, blinding of participants and personnel, blinding of outcome assessment, incomplete outcome data, selective outcome reporting, and other bias”.

### 2.5. Data Analysis

The blood lipid levels of different units were uniformly converted to mg/dL (For TC, LDL-C, and HDL-C, 1 mmol/L was converted to 38.7 mg/dL; for TG, 1 mmol/L was converted to 88.5 mg/dL). Excel 2010 was used for data collection and processing, and Stata SE 15.0 was used for data analysis.

The Mean changes (SD) of the outcome measures (TC, LDL-C, HDL-C, TG) were used to calculate the overall effect size. The standard deviation (SD) of each outcome was calculated by the variables before and after intervention according to the formula as follows: SD_change_ = (SD^2^
_baseline_ + SD^2^ _endpoint−2_ × R × SD_baseline_ × SD_endpoint_) 1/2, correlation coefficient R = 0.5 [33].

In addition, heterogeneity was evaluated via *I*^2^ metrics and chi-squared statistics. The fixed model was generally used, and the random effect model was applied only when *I*^2^ > 50% or *p*-value of χ^2^ test < 0.10, indicating high heterogeneity. Additionally, subgroup analyses were conducted based on the resveratrol dose, population condition and duration of trials.

To assess the influence of a single study on the overall results, sensitivity analysis was used in our study. We applied Funnel plots and Egger’s test to identify potential publication bias. The trim and fill analysis (metatrim) were used to assess the effect of publication bias on the stability of the results. We also executed fractional polynomial modeling (polynomials) to explore the potential non-linear effects of resveratrol dosage (mg) and duration of treatment (weeks) [34].

## 3. Results

### 3.1. Baseline Characteristics of Included Studies

A total of 3510 articles were retrieved from four databases. After removing 427 duplications and 3083 irrelevant articles based on the title and abstract, 61 trials were included for eligibility assessment. An additional 44 records were excluded for the following reasons: unavailable full text, no randomized controlled trials, data scarcity, and no appropriate control group. Finally, 17 studies were included in this meta-analysis. Of note, the data from one article was divided into two trials, in which resveratrol was consumed at a dosage of 100 mg/d for 48 weeks followed by a dosage of 200 mg/d for 48 weeks. Therefore, 17 trials with 18 data sets were included in the meta-analysis. The complete flow diagram of the selected studies is shown in Figure 1.

### 3.2. Study Characteristics

The main characteristics of the included studies are outlined in Table 1. Totally, there were 968 subjects included in these published studies from 2010 to 2019. The included studies were conducted in Asia (7 trials), Europe (6 trials), Australia (1 trial), and America (3 trials). The age range was from 18 to 85 years. The duration of resveratrol intake varied from 4 to 48 weeks and the intervention dosage ranged from 10 to 3000 mg/day.

### 3.3. Results of Meta-Analysis

In total, 17 publications with 18 effect sizes were included in the analysis of the TC level. The intervention of resveratrol exerted a statistically significant effect on the level of serum TC (Mean difference = −10.28; 95%CI: −13.79, −6.76, *p* < 0.001, *I*^2^ = 26.7%, Figure 2A). With regard to the non-linear dose–response analysis, a significant effect of duration of resveratrol supplementation (p-nonlinearity = 0.321, Figure 3A) and resveratrol dosage (p-nonlinearity = 0.556, Figure 4A) on the TC level was not present.

Among the 17 effect sizes from 16 studies for the HDL-C analysis, it was found that there was no significant effect of resveratrol supplementation on the HDL-C level (Mean difference = 1.36; 95%CI: −0.06, 2.79, *p* = 0.061, *I*^2^ = 38.5%, Figure 2B). In addition, there was no significant nonlinear relationship with dosage (p-nonlinearity = 0.060, Figure 3B) or duration (Pnon-linearity = 0.271, Figure 4B) on HDL-C.

The results of serum LDL-C were calculated in 16 comparisons from 15 studies. The applied random-effects model showed that resveratrol intake significantly decreased the LDL-C level (Mean difference = −5.69; 95%CI: −11.07, −0.31, *p* = 0.038, *I*^2^ = 64.2%, Figure 2C). In the non-linear dose–response analysis, we observed a significant effect of supplementation dosage (p-nonlinearity = 0.002, Figure 3C) but did not observe a significant effect of duration of intervention on LDL-C (p-nonlinearity = 0.415, Figure 4C).

Pooled effect size based on 17 studies (with 18 effect sizes), indicated that resveratrol supplementation significantly reduced the TG level (Mean difference = −8.56; 95%CI: −12.37, −4.75, *p* < 0.001, *I*^2^ = 0.0%, Figure 2D). Furthermore, the effect of resveratrol dosage (Pnon-linearity = 0.062, Figure 3D) or duration (Pnon-linearity = 0.416, Figure 4D) on the TG level was insignificant.

### 3.4. Subgroup Analysis

Since only the heterogeneity of LDL-C was more than 50%, a subgroup analysis of LDL-C was conducted to identify the source of heterogeneity, which is shown in Table 2. In the subgroup analysis by the treatment dosage (*I*^2^ = 20.5%, *p* = 0.267), duration of the intervention (*I*^2^ = 36.2%, *p* = 0.152), study population (Obese/Overweight: *I*^2^ = 0.0%, *p* = 0.449 and Stroke: *I*^2^ = 0.0%, *p* = 0.887 and other: *I*^2^ = 0.0%, *p* = 0.911), between-study heterogeneity disappeared. In these analyses, a significant alteration in LDL-C was observed in trials with ≥500 mg/d resveratrol (Mean difference = 2.44; 95%CI: −8.21,13.08, *p* = 0.004), those with a duration of ≥12 weeks (Mean difference = −6.01; 95%CI: −13.26,1.25, *p* < 0.001), those performed in subjects with T2DM (Mean difference= −4.21; 95%CI: −18.78,10.36, *p* < 0.001).

### 3.5. Publication Bias and Sensitivity Analysis

The black dots in the funnel plot represent the studies that were included. The larger the sample size, the smaller the standard error and the higher the distribution. Conversely, the smaller the sample size, the larger the standard error and the higher the distribution.As shown in Figure 5, there was no evidence for publication bias except for HDL-C (total cholesterol Egger’s test: *p* = 0.134, LDL-C Egger’s test: *p* = 0.115, HDL-C Egger’s test: *p* = 0.003 and triglyceride Egger’s test: *p* = 0.104) via funnel plots and Egger’s tests. The trim and fill analysis (metatrim) that assesses the effect of publication bias on the stability of the results showed that there was no trimming performed and data were unchanged (detailed in Figure 6). The small circles in the figure represent roughly the same idea as the funnel plot above.Moreover, the *p* values before and after the trim and fill analysis were less than 0.05, indicating that the results were robust. Therefore, although there was a publication bias in HDL-C data, it exerted an insignificant impact on the stability of the results. Additionally, results from the sensitivity analyses indicated that there was no substantial change in the result of the TC, LDL-C, HDL-C, or TG level.

## 4. Discussion

In this meta-analysis, we found that the intervention of resveratrol affected the levels of TC, LDL-C, and TG significantly, but not the level of HDL-C. In the nonlinear dose–response analysis, we observed a significant effect of supplementation dosage on the level of LDL-C. In addition, the reduction of LDL-C was more significantly in the trials with a duration ≥12 weeks and in subjects with T2DM.

Although a previous meta-analysis showed a non-significant hypolipidemic effect of the resveratrol supplementation, which might be due to the limited number of trials included [31], as we collected and synthesized the most recent evidence, it was found that the supplementation of resveratrol exerted significant effects on reducing serum TC and TG. As documented, four metabolic pathways including triacylglycerol metabolism and fat accumulation in tissues, fatty acid uptake from circulating triacylglycerol, de novo adipogenesis, as well as lipolysis, and fatty acid oxidation involved in the regulation of TC and TG by resveratrol [42]. The underlying mechanisms include: (1) upregulating the expressions of reverse cholesterol transporters, such as PPARc, LXRa, 27-hydroxylase, and ABCA1, which promote cholesterol spillover and thus prevent cholesterol accumulation; (2) increasing the apolipoprotein (APO) A-I/apoB ratio by down-regulating 3-hydroxy-3-methylglutaryl-coA reductase, which is negatively correlated with cardiovascular risk.; (3) regulating the SIRT1-PPARγ pathway and its downstream genes FAS and ACC [43]. Given the regulatory role of resveratrol in controlling TC and TG, the supplementation of resveratrol has been implicated in a number of diseases associated with abnormal lipid metabolism, such as non-alcoholic fatty liver disease, coronary heart disease, and atherosclerosis [44].

In contrast to TC and TG, our meta-analysis did not show a significant effect of resveratrol on HDL-C, which is inconsistent with previous reports [20,37]. Such inconsistency may be resulted from the population in these two articles and the influence of other independent factors on the experiment. In addition, the potential mechanism of resveratrol in regulating HDL-C, if any, has not been explored. Hereby, the effect of resveratrol supplementation on HDL-C and the associated mechanisms are worthwhile for future research.

LDL-C was found to be reduced by resveratrol supplementation in this meta-analysis. The resveratrol-induced reduction of LDL-C was associated with the antioxidant effect of resveratrol. The LDL oxidation could be inhibited by reducing the electrophoretic mobility, followed by blocking the internalization of the oxidized lipoprotein [45,46].

In this meta-analysis, we conducted a dose-response analysis and found a nonlinear relationship between the dosage of resveratrol and the level of LDL-C, which indicates that the effect of resveratrol on LDL-C is dependent on the intervention dose. Surprisingly, in the following subgroup analysis, we found that a high dosage of resveratrol (≥500 mg/d) showed an opposite effect size, which is in line with the upward trend we observed in the dose-response plot wherever the dose is above 500 mg/d. Such a finding suggests that the dosage is a crucial factor in the resveratrol supplementation. In addition to LDL-C, the opposite effects of a high dose of resveratrol were reported in other measurements. Body weight and BMI might be increased by resveratrol supplementation with a dosage more than 500 mg/d, which was different from the effect of the lower dosage [47]. It was also reported that resveratrol had significant antiapoptotic and cardioprotective effects, while it suppressed cardiac functions at higher dosages (25 and 50 mg/kg) [48]. Additionally, a low resveratrol dose (5 mg/day) suppressed intestinal adenoma development more potently than did the higher dose (1 g/day) [49]. Collectively, the dosage is an essential factor of resveratrol supplementation, which may result in opposite outcomes of the intervention. Hereby, the dosage of resveratrol should be considered carefully in the design of clinical trials. The differential regulatory roles of resveratrol at different dosages need to be further investigated in future studies.

Of note, the supplementation of resveratrol showed a more pronounced effect on reducing LDL-C in subjects with T2DM. It has been reported by other studies that intervention of resveratrol could reduce the TG level [30], mitigating insulin resistance, lower fasting blood glucose, and ameliorate oxidative stress [28] in T2DM patients, which implicated a broad range of benefits of resveratrol in T2DM subjects. Of note, the summarized results showed no significant incidence of adverse effects of resveratrol in T2DM subjects [28,50].

It has to be admitted that the current meta-analysis has several limitations. Firstly, both the dosage and duration of the intervention varied among the included studies. The results of the subgroup analysis showed that the studies using low-dose resveratrol had a significant effect on lowering blood lipids, whereas the studies using high-dose resveratrol exerted the opposite effect. Of note, the duration of the intervention with low dosage was longer. Hence, the cross-talk of the effects of the dosage and duration of resveratrol on blood lipid profile need to be further discussed. Secondly, whether gender difference exists in response to the resveratrol intervention is unclear, since three all-male trials were included but there was no trial with only female subjects. Additionally, there is a publication bias in the HDL-C data, although the publication bias had no significant impact on the stability of the results through the analysis of the cut-and-fill method. Finally, since subjects in some trials are patients with T2DM, obesity, NAFLD or stroke, the supplementation of resveratrol might be used with medical drugs. Thus, it is possible that the effects of resveratrol could be affected by other drugs, which would require more specific studies.

## 5. Conclusions

In conclusion, this meta-analysis indicated that the supplementation of resveratrol could significantly affect the serum levels of TC, LDL-C and TG, but not the level of HDL-C. In addition, the dosage of the resveratrol intervention is an essential factor that should be considered cautiously. Findings from this meta-analysis could be helpful for providing suggestions for the use of resveratrol as nutraceutical.

## Figures and Tables

**Figure 1 nutrients-14-03755-f001:**
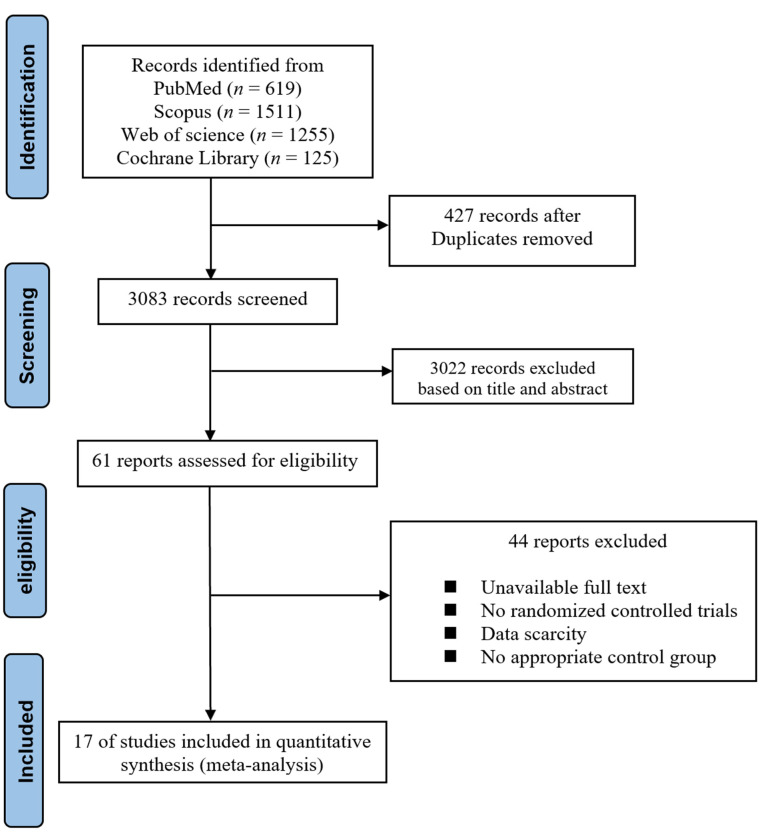
PRISMA flow chart of selected trials.

**Figure 2 nutrients-14-03755-f002:**
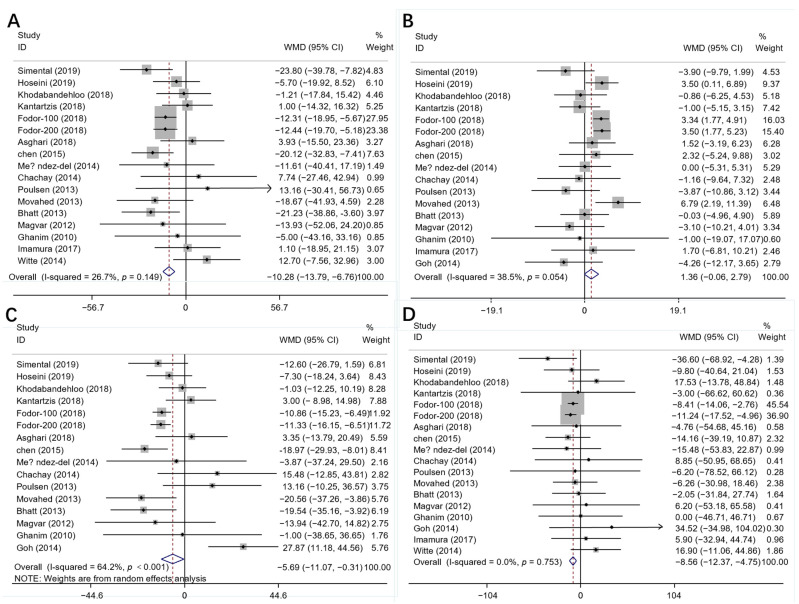
The effects of resveratrol on (**A**) TC, (**B**) HDL-C, (**C**) LDL-C and (**D**) TG. TC = Total cholesterol; LDL-C = Low-density lipoprotein cholesterol; HDL-C = High-density lipoprotein cholesterol; TG = Triglyceride; WMD = Weighted mean difference [19,20,21,22,23,24,25,26,27,32,35,36,37,38,39,40,41].

**Figure 3 nutrients-14-03755-f003:**
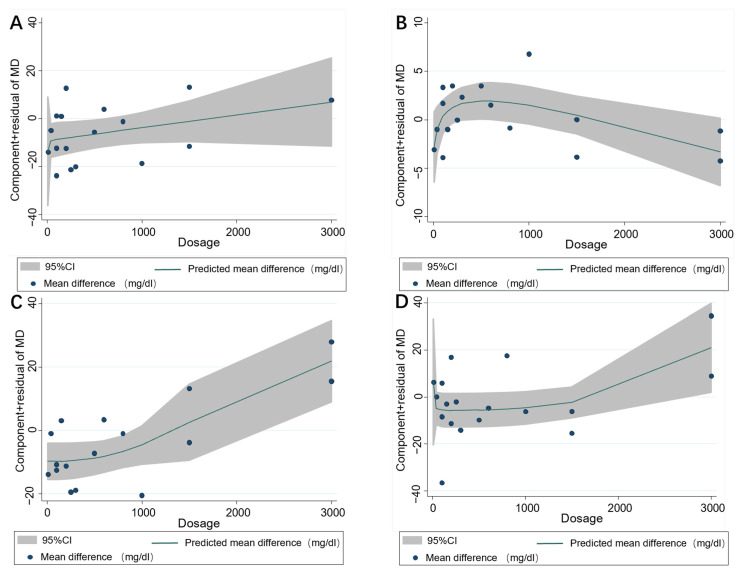
Nonlinear dose–response relations between resveratrol dosage (mg) and unstandardized mean difference (mg/dL) in (**A**) TC, (**B**) HDL-C, (**C**) LDL-C and (**D**) TG. The 95% CI is demonstrated in the shaded regions. TC = Total cholesterol; LDL-C = Low-density lipoprotein cholesterol; HDL-C = High-density lipoprotein cholesterol; TG = Triglyceride.

**Figure 4 nutrients-14-03755-f004:**
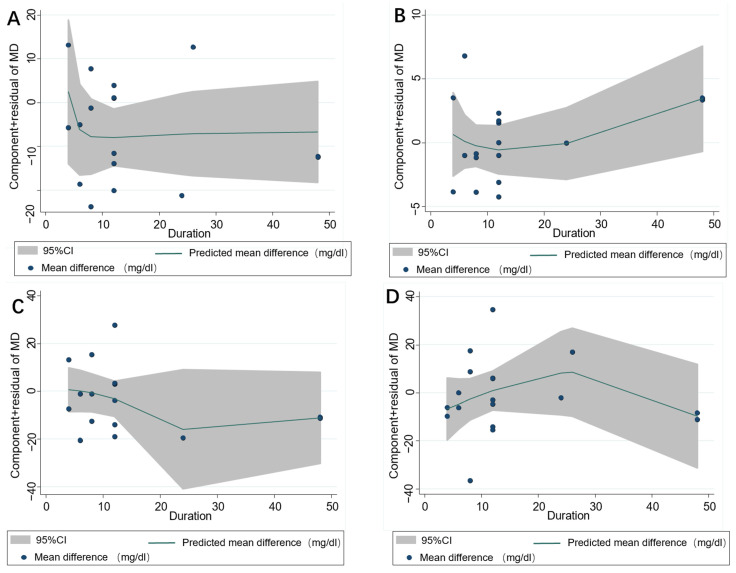
Nonlinear dose–response relations between resveratrol duration (week) and unstandardized mean difference (mg/dL) in (**A**) TC, (**B**) HDL-C, (**C**) LDL-C and (**D**) TG. The 95% CI is demonstrated in the shaded regions. TC = Total cholesterol; HDL-C = High-density lipoprotein cholesterol; LDL-C = Low-density lipoprotein cholesterol; TG = Triglyceride.

**Figure 5 nutrients-14-03755-f005:**
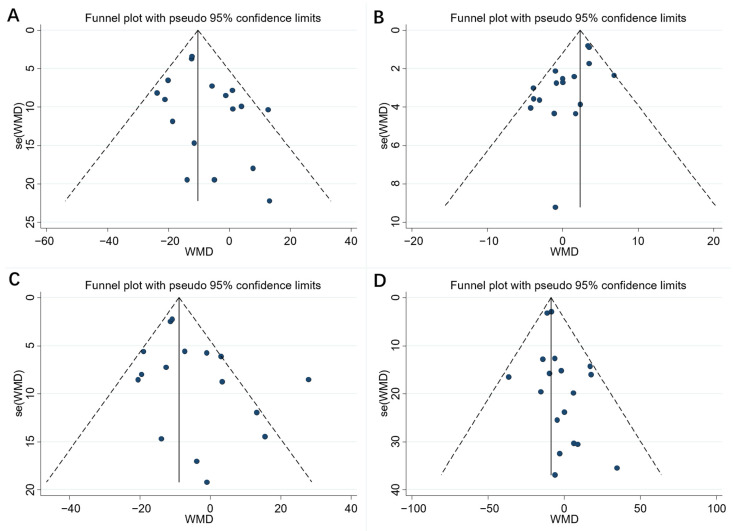
Funnel plots to evaluate publication bias, and effect of resveratrol for (**A**) TC Egger’s test (*p* = 0.134), (**B**) HDL-C Egger’s test (*p* = 0.03), (**C**) LDLC Egger’s test (*p* = 0.115), and (**D**) TG Egger’s test (*p* = 0.104). A = TC; B = LDL-C; C = HDL-C; D = TG. TC = Total cholesterol; HDL-C = High-density lipoprotein cholesterol; LDL-C = Low-density lipoprotein cholesterol; TG = Triglyceride. The black dots in the funnel plot represent the studies that were included. The larger the sample size, the smaller the standard error and the higher the distribution. Conversely, the smaller the sample size, the larger the standard error and the higher the distribution.

**Figure 6 nutrients-14-03755-f006:**
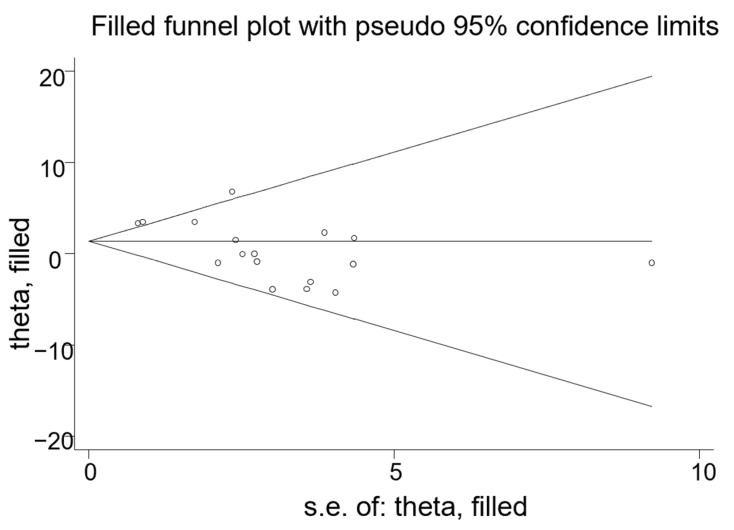
Filled funnel plot of effect estimate by standard error in means for HDL-C. The dots in the funnel plot represent the studies that were included. The larger the sample size, the smaller the standard error and the higher the distribution. Conversely, the smaller the sample size, the larger the standard error and the higher the distribution.

**Table 1 nutrients-14-03755-t001:** Characteristics of the studies included.

Literature	Year	Study Region	Participant	Female%	Age Range	Sample (I/C)	Circle	Dose (mg/d)	Study Design
Simental [19]	2019	Mexico	Dyslipidemia	74.2	20–65	31/31	8 week	100	parallel
Hoseini [20]	2019	Iran	T2DM and coronary heart disease (CHD)	NA	40–85	28/28	4 week	500	parallel
Khodabandehloo [24]	2018	Iran	T2DM	51.1	30–70	25/20	8 week	800	parallel
Kantartzis [35]	2018	Germany	BMI ≥ 27	NA	18–70	53/52	12 week	150	parallel
Fodor-100 [32]	2018	Romania	Stroke	60.5	≥55	81/92	48 week	100	parallel
Fodor-200 [32]	2018	Romania	Stroke	60.5	≥55	55/92	48 week	200	parallel
Asghari [25]	2018	Iran	NAFLD	33.3	20–60	30/30	12 week	600	parallel
Chen [21]	2015	China	NAFLD	30	20–60	30/30	12 week	300	parallel
Me’ ndez-del [26]	2014	Mexico	Metabolic syndrome	NA	30–50	11/10	12 week	1500	parallel
Chachay [27]	2014	Australia	NAFLD	0	36–61	10/10	8 week	3000	parallel
Poulsen [36]	2013	Denmark	Obese Men	0	18–70	12/12	4 week	1500	parallel
Movahed [37]	2013	Iran	T2DM	50	45–59	33/31	6 week (45 day)	1000	parallel
Bhatt [38]	2013	India	T2DM	63.2	30–70	28/29	24 week (6 month)	250	parallel
Magvar [23]	2012	Hungary	Stable coronary artery disease	35	42–80	20/20	12 week	10	parallel
Ghanim [39]	2010	America	Healthy	NA	31–41	10/10	6 week	40	parallel
Imamura [40]	2017	Finland	T2DM	48	47–68	25/25	12 week	100	parallel
Witte [41]	2014	Germany	Overweight	39.1	50–75	23/23	26 week	200	parallel
Goh [22]	2014	Singapore	T2DM	0	40–69	5/5	12 week	3000	parallel

N/A = Not Available.

**Table 2 nutrients-14-03755-t002:** The results of subgroup analysis in included studies.

Index	Subgroup	No. of Trials	Mean Difference		*p*	I^2^ (%)	*p* Value of Heterogeneity
			Mean	95%CI			
LDL-C	Resveratrol dose (g/day)						
	≥500 mg	8	2.44	−8.21, 13.08	0.004	66.4	0.654
	<500 mg	8	−11.16	−14.94, −7.38	0.267	20.5	0.001
	Population condition						
	T2DM	5	−4.21	−18.78, 10.36	0.001	82	0.571
	NAFLD	3	−2.77	−23.22, 17.69	0.018	75	0.791
	Obese/Overweight	2	5.11	−0.86, −0.18	0.449	0	0.348
	Stroke	2	−11.07	−14.31, −7.84	0.887	0	0.001
	Others	4	−10.75	−22.09, 0.59	0.911	0	0.063
	Duration of the trial						
	≥12 weeks	9	−6.01	−13.26, 1.25	0.001	73.4	0.105
	<12 weeks	7	−5.12	−13.13, 2.89	0.152	36.2	0.210

## Data Availability

Not applicable.

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
