# Peer review of "The Effect of Resveratrol on Blood Lipid Profile: A Dose-Response Meta-Analysis of Randomized Controlled Trials"

_nutrients, 2022, doi:10.3390/nu14183755_

Round 1

Reviewer 1 Report

The article “The effect of resveratrol on blood lipid profile: a dose-response meta-analysis of randomized controlled trials” by X. Cao and colleagues performed a meta-analysis of researches about resveratrol supplementation and its effect on the TC, TG, LDL, and HDL blood levels.

The study is interesting and carried out very-well in order to obtain significant interpretation of results that are often contrasting among them.

In my opinion, the article does not present major issues and deserves publication in Nutrients.

Minor issues are listed below.

The use of unexplained acronyms (although widely used) in the abstract is strongly discouraged.

Many spaces are missing between words, numbers and units, before references, etc.

“cis” and “trans” must be in italics.

Line 30: correct “Eu” to “EU”.

Line 32: correct to “3,5,4’-trihydroxystilbene”.

Line 104: I suppose it is “mg/dL”.

Letters (A, B, C, D) are missing in the caption of Figure 2.

Reviewer 2 Report

-The rationale for the study is unclear as the introduction is a bit confusing, comprising several pieces of apparently unlinked information. A more integrated appraisal of the relevant literature would be appropriate to provide the context for the study.

-The following recent pertinent reports should be mentioned:

PMID: 33335015

PMID: 29407880

PMID: 31316465

PMID: 27385446

PMID: 35895130

-The strengths and limitations of the study should be deeply addressed, taking into account sources of potential bias or imprecision: Discuss both direction and magnitude of any potential bias.

-It is advisable to the Authors to incorporate a pictorial or cartoon representation of the main results of the study to increase the overall impact of the manuscript.

Reviewer 3 Report

The manuscript can be of interest to wide readers of Journals and contributes to existing knowledge on the subject matter. However, I have pointed out few pertinent points for improving the clarity of the content and boosting the scientific soundness of the manuscript.

Abstract: Define all the abbreviated forms at their first use.

Introduction

More information may be added on pertinence of importance of Resveratrol.

Include citations to the following

“Although a number of studies have reported the effects of resveratrol on blood lipids in humans, the results are inconsistent”

Line 185: Avoid using references in the Result sectionThe manuscript can be of interest to wide readers of Journals and contributes to existing knowledge on the subject matter. However, I have pointed out few pertinent points for improving the clarity of the content and boosting the scientific soundness of the manuscript.

Abstract: Define all the abbreviated forms at their first use.

Introduction

More information may be added on pertinence of importance of Resveratrol.

Include citations to the following

“Although a number of studies have reported the effects of resveratrol on blood lipids in humans, the results are inconsistent”

Line 185: Avoid using references in the Result section

Round 2

Reviewer 2 Report

-